

# A new method for inferring city emissions and lifetimes of nitrogen oxides from high-resolution nitrogen dioxide observations: A model study

Fei Liu[1,2], Zhining Tao[1,2], Steffen Beirle[3], Joanna Joiner[2], Yasuko Yoshida[2,4], Steven J. Smith[5], K. Emma Knowland[1,2], Thomas Wagner[3]

[1]Universities Space Research Association (USRA), Goddard Earth Sciences Technology and Research (GESTAR), Columbia, MD, 21046, USA
[2]NASA Goddard Space Flight Center, Greenbelt, MD, 20771, USA
[3]Max-Planck-Institut für Chemie, Mainz, 55128, Germany
[4]Science Systems and Applications Inc., Lanham, MD, 20706, USA
[5]Joint Global Change Research Institute, Pacific Northwest National Laboratory, College Park, MD, 20740, USA

*Correspondence to*: Fei Liu (fei.liu@nasa.gov)

## Abstract

We present a new method to infer nitrogen oxides ($NO_x$) emissions and lifetimes based on tropospheric nitrogen dioxide ($NO_2$) observations together with reanalysis wind fields for cities located in polluted backgrounds. Since the accuracy of the method is difficult to assess due to lack of "true values" that can be used as a benchmark, we apply the method to synthetic $NO_2$ observations derived from the NASA-Unified Weather Research and Forecasting (NU-WRF) model at a high horizontal spatial resolution of 4 km × 4 km for cities over the continental US. We compare the inferred emissions and lifetimes with the values given by the NU-WRF model to evaluate the method. The method is applicable to 26 US cities. The derived results are generally in good agreement with the values given by the model, with the relative differences of 2 % ± 17 % (mean ± standard deviation) and 15 % ± 25 % for lifetimes and emissions, respectively. Our investigation suggests that the use of wind data prior to satellite overpass time improves the performance of the method. The correlation coefficients between inferred and NU-WRF lifetimes increase from 0.56 to 0.79 and for emissions increase from 0.88 to 0.96 when comparing results based on wind fields sampled simultaneously with satellite observations and averaged over 9 hours data prior to satellite observations, respectively. We estimate that uncertainties in $NO_x$ lifetime and emissions arising from the method are approximately 15 % and 20 %, respectively, for typical (US) cities. We expect this new method to be applicable to $NO_2$ observations from the TROPOspheric Monitoring Instrument (TROPOMI) and geostationary satellites, such as Geostationary Environment Monitoring Spectrometer (GEMS) or the Tropospheric Emissions: Monitoring Pollution (TEMPO) instrument, to estimate urban $NO_x$ emissions and lifetimes globally.



## 1 Introduction

Nitrogen oxides (NO$_x$), consisting of nitrogen dioxide (NO$_2$) and nitric oxide (NO), are important atmospheric trace gases that actively participate in the formation of tropospheric ozone and secondary aerosols and accordingly have a significant effect on human health and climate (Seinfeld and Pandis, 2006). NO$_x$ emission sources include anthropogenic activities, biomass burning, soil emissions, and lightning. Fossil-fuel burning from mobile and industrial emitters represents the largest source of anthropogenic NO$_x$ emissions; these sources are usually clustered near densely populated urban areas (Crippa et al., 2018).

We traditionally rely on a bottom-up method to estimate anthropogenic NO$_x$ emissions for a country or a region based on their total fuel use and averaged emission factors, which are subject to uncertainties due to incomplete understanding of real world operating conditions and spatial distributions (Butler et al., 2008). Some sources may be missing from bottom-up emission inventories (McLinden et al., 2016). Additionally, estimates of NO$_x$ emissions may become outdated when fuel consumption and emission factors change dramatically. For instance, NO$_x$ emissions from China decreased by 21 % from 2011 to 2015 due to wide deployment of denitration devices (Liu et al., 2016a). Inferring emissions for individual cities is even more challenging, due to the difficulties in acquiring a complete and reliable database for fuel consumptions and emission factors at city level. Proxies such as population density, industrial productivity, and road network maps are often used to downscale national/regional emissions to finer scales, which may incorrectly allocate emission sources spatially (Butler et al., 2008).

Satellite observations of tropospheric NO$_2$ have been widely used to infer the strength of NO$_x$ emissions. Satellite instruments, e.g., the Ozone Monitoring Instrument (OMI; Levelt et al., 2006, 2018) and TROPOspheric Monitoring Instrument (TROPOMI; Veefkind et al., 2012), are used to retrieve the column density of NO$_2$ in a vertical column of air. These data can then be related to NO$_x$ emissions by considering chemical conversion and transport. Chemical transport models (CTMs) were initially employed to exploit NO$_2$ measured from space as a constraint to improve NO$_x$ emission inventories based on mass balance (e.g., Martin et al., 2003; Kim et al., 2009; Lamsal et al., 2011). Techniques such as the four-dimensional variational (4D-Var) method (e.g., Henze et al., 2007, 2009), extended Kalman filter (e.g., Ding et al., 2017), ensemble Kalman filter (e.g., Miyazaki et al., 2017), and hybrid mass balance/4D-Var (e.g., Qu et al., 2019) have also been used to improve emissions estimates within CTMs.

Several studies have inferred emissions independent of CTMs (e.g., Beirle et al., 2011; Liu et al., 2017; Laughner and Cohen, 2019). Such investigations were inspired by a pioneering study that used the downwind decay of NO$_2$ in continental outflow regions to estimate the global NO$_x$ lifetime and total emissions (Leue et al., 2001). Beirle et al. (2011) first proposed an empirical function to describe the plume distribution around an isolated city without inputs from CTMs. Follow-up studies have adopted this function to provide estimates of NO$_x$ emissions from power plants and cities based on OMI (e.g., de Foy et al., 2015 and Lu et al., 2015) and TROPOMI (e.g., Goldberg et al., 2019) observations. Additional methods, such as the plume rotation technique (Pommier et al., 2013; Valin et al., 2013) and the divergence approach (Beirle et al., 2019), were developed to refine the approach of Beirle et al. (2011). More recent studies explored additional constraints for the empirical function using simulated atmospheric composition from models (e.g., Lorente et al., 2019; Lange et al., 2021). For sources with a polluted background, Liu et al. (2016b) proposed a different fitting function to consider the interferences from surrounding sources; this approach has been used to estimate NO$_x$ emissions for European cities (Verstraeten et al., 2018).

The uncertainties in satellite-derived emissions inferred from CTM-independent approaches have rarely been investigated. Existing studies usually quantify the uncertainties based on results from sensitivity analyses (e.g., Beirle et al., 2011), since we usually lack "true values" that can be used as a benchmark for validation. de Foy et al. (2014) tested CTM-independent approaches using simulated NO$_2$ column densities from a single point source with a specified emission and chemical lifetime. The good consistency



between the derived values and the specified values given to drive the simulation suggests that the uncertainty of the Beirle et al. (2011) approach is small for an ideal, isolated source. However, the performance of the approach in the real world with complex source distributions has not yet been evaluated.

Here, on the basis of previous approaches (Beirle et al., 2011; Liu et al., 2016b), we develop a new CTM-independent approach

for inferring $NO_x$ lifetimes and emissions for cities with polluted backgrounds and complex spatial distribution of interfering emissions. We use synthetic $NO_2$ observations derived from a model simulation to evaluate the performance of the new approach and to estimate its uncertainties. An overview of the synthetic observations, the methodology and features of the new CTM-independent approach is provided in Sect. 2. We evaluate results by comparing the inferred emissions and lifetimes with values from the model simulation in Sect. 3.1. Section 3.2 compares the performance of the method developed in this work with previous

approaches (Beirle et al., 2011; Liu et al., 2016b). Section 3.3 discusses the uncertainties of $NO_x$ lifetimes and emissions derived from the new approach. Section 4 presents a summary of the performance of the new method and the future work plans for applying the method to satellite observations.

## 2 Data and method

In this section, we develop an evaluation system to assess the performance of our newly-developed CTM-Independent SATellite-

derived Emission estimation Algorithm for Mixed-sources (MISATEAM). Figure 1 displays the schematic of the evaluation system. MISATEAM uses satellite retrievals of tropospheric $NO_2$ vertical column densities (VCDs), together with wind information from a meteorological reanalysis, to infer $NO_x$ lifetimes and emissions for cities. Cities are usually non-isolated sources with polluted backgrounds (Fig. S1). Additionally, emissions from cities may spread out and make cities not (quasi) point sources even at the footprint of satellite observations (a few km; Fig. 3). We refer to these cities as mixed-sources.

To evaluate MISATEAM, we replace satellite observations with synthetic $NO_2$ VCDs derived from the NASA-Unified Weather Research and Forecasting (NU-WRF) model (Tao et al., 2013; Peters-Lidard et al., 2015) (Sect. 2.1). We then apply MISATEAM to the synthetic $NO_2$ VCDs and NU-WRF meteorological fields to infer urban $NO_x$ lifetimes and emissions (Sect. 2.2). We investigate the impact of temporal variations in wind fields on derived $NO_x$ lifetimes and emissions (Sect. 2.3). In Section 2.4, we describe the benchmark $NO_x$ emissions directly given by NU-WRF and $NO_x$ lifetimes deduced from known $NO_x$ emissions and

concentrations (hereafter referred to as "given emissions and NU-WRF lifetimes") that we will compare with the MISATEAM-derived lifetimes and emissions. Analysis of the uncertainties in these datasets, including satellite observations and wind fields, is outside the scope of the study. We briefly discuss the potential impact of ignoring systematic errors in Sect. 3.3.2.

### 2.1 Synthetic $NO_2$ VCDs: NU-WRF simulations

We use a regional modelling system, NU-WRF (Tao et al., 2013; Peters-Lidard et al., 2015), to simulate tropospheric $NO_2$ VCDs

over the continental US. NU-WRF was developed from the advanced research versions of WRF (Michalakes et al., 2001) and WRF-Chem (Grell et al., 2005) with the addition of several NASA-developed components (e.g., Chou and Suarez, 1999; Chin et al., 2002, 2007; Kumar et al., 2006; Peters-Lidard et al., 2007; Shi et al., 2010). The gas-phase chemical mechanism in NU-WRF is the second-generation regional acid deposition model (RADM2, Gross and Stockwell, 2003). The aerosol module is the Goddard Chemistry Aerosol Radiation and Transport (GOCART) model (Chin et al., 2002). We use the anthropogenic emissions based on

the 2011 National Emissions Inventory (NEI) compiled by the US Environmental Protection Agency (US EPA, NEI 2011) but with a few modifications, in which the measurements from OMI, the ground-based Air Quality System (AQS), the in-situ continuous emissions monitoring in power plants, and the Air Pollutant Emissions Trends Data compiled by the US EPA





(https://www.epa.gov/air-emissions-inventories/air-pollutant-emissions-trends-data), have been employed to adjust the baseline emissions (Tong et al., 2015; Tao et al., 2020). The simulation also includes the fire emissions from the Global Fire Data version

4 with small fires (GFED v4s, van der Werf et al., 2017; Randerson et al., 2015); biogenic emissions from the online calculation using the Model of Emissions of Gases and Aerosols from Nature version 2 (MEGAN2, Guenther et al., 2006); dust emissions from the on-line estimation based on the surface wind speed, soil moisture, and soil erodibility (Ginoux et al., 2001; Kim et et al., 2017); and sea salt emissions from the on-line computation based on the method by Gong (2003).

We run NU-WRF for 2016 at a high horizontal spatial resolution of 4 km × 4 km and 40 vertical layers extending from surface to

50 hPa in this study. We integrate $NO_2$ concentrations from the surface to the tropopause (~10 km) to calculate tropospheric $NO_2$ VCDs. The meteorological initial and lateral boundary conditions are derived from the Modern Era Retrospective-Analysis for Research and Applications version 2 (MERRA-2, Rienecker et al., 2011; Gelaro et al., 2017). The chemical initial and lateral boundary conditions are derived from the results of the Community Atmosphere Model with chemistry (CAM-chem, Lamarque et al., 2012). A 7-day model spin up following the recommendation by Berge et al. (2001) is used.

Figure 2A illustrates the six-month average of the simulated hourly mean tropospheric $NO_2$ VCDs sampled at the local overpass time of TROPOMI. The $NO_x$ emissions used to drive NU-WRF over the model domain for the same time period are presented in Fig. 2B. We focus on cities with populations > 200,000, which have been defined as medium-size urban areas in Organisation for Economic Co-operation and Development (OECD) countries. Nearby cities (located within 50 km of the largest city in a given urban area) are considered as one city cluster when applying MISATEAM to infer lifetimes and emissions. Cities on the boundary

of the model domain, e.g., Seattle and San Francisco, are excluded from the following analysis, because the data for their inflow/outflow plumes are partially missing from the model output and thus do not meet the requirements of MISATEAM (see details of the fit interval in Sect. 2.2). This filtering results in a total of 60 cities and urban conglomerations as the candidates for applying MISATEAM.

### 2.2 Emission estimation algorithm

We develop MISATEAM based on the methods of Beirle et al. (2011) and Liu et al. (2016b). We adapt the basic approach of Liu et al. (2016b) that estimates $NO_2$ spatial emissions patterns, $E(x)$, using $NO_2$ observations, $LD_{calm}(x)$, following

$$E(x) = \frac{ratio \times [LD_{calm}(x) - b]}{\tau},$$   (1)

where $E(x)$ is a function of distance from the city center (denoted by $x$) in a particular direction and integrated over a given distance in a direction $y$ (perpendicular to that of $x$). The mean emission maps (two-dimensional, 2D) are reduced to 1D along the respective

direction $x$ by integration across the direction $y$. $LD_{calm}(x)$ are the so-called $NO_2$ line densities, defined as the observed $NO_2$ VCDs (units molec $cm^{-2}$) under calm wind conditions (wind speed < 2 m $s^{-1}$) integrated in the same way as $E(x)$ to give units of molec $cm^{-1}$ as in Beirle et al. (2011). $ratio$ is the ratio of $NO_x$ to $NO_2$. $b$ represents a $NO_2$ background, which is derived by analyzing the distribution of $NO_2$ VCDs. $\tau$ is the $NO_x$ lifetime, which is the fitting parameter. We then use the following model function, $f(x)$, to describe $NO_2$ line densities under windy conditions (wind speed > 2 m $s^{-1}$) $LD_{windy}(x)$:

$$f(x) = \frac{E(x)}{ratio \times w} * e^{-\frac{x}{w \times \tau}} + b$$

$$= \frac{[LD_{calm}(x) - b]}{\tau \times w} * e^{-\frac{x}{w \times \tau}} + b,$$   (2)

where $w$ is the mean wind speed at the emission level in a given direction $x$, and * denotes convolution. Figure 3 illustrates the calculation of $LD_{windy}(x)$.





Note that $\tau$ is assumed to be an effective mean dispersion lifetime, i.e., the result of the effect of deposition, chemical conversion,
and wind advection, because we do not consider downwind changes in the fitting function, such as due to variations in $w$ or $ratio$
or lifetime itself. Additional technical details of the model function $f(x)$ and its differences compared to those proposed by Liu et
al. (2016b) are given in Appendix A.

To estimate $b$, we first calculate the mean $NO_2$ VCD under calm wind conditions for grid cells within the lowest 5th percentile of
$NO_2$ VCDs for each city. This produces a good approximation of the mean $NO_2$ VCD for grid cells with low $NO_x$ emissions (i.e.,
the lowest 5th percentile of $NO_x$ emissions). We then multiply this mean value by the width of the across-wind integration interval
to derive $b$.

We use $ratio$ of 1.32 to represent "typical urban conditions and noontime sun" (Seinfeld and Pandis, 2006). We investigate the
effect of using a constant value of $ratio$ on derived emissions in Sect. 3.1; it is found to be insignificant. Note that the derived
lifetime $\tau$ is not sensitive to the magnitude of $ratio$, as $\tau$ is determined by the relative decay pattern.

Finally, we use estimates of $b$ and $ratio$ in Eq. (2), along with values of $w$, $LD_{calm}(x)$, and $LD_{windy}(x)$ from the model simulation to
infer $\tau$ and $E(x)$. As displayed in Fig. 1, we use the NU-WRF high-resolution tropospheric $NO_2$ VCDs sampled at the local overpass
time of TROPOMI as the synthetic $NO_2$ VCD observations, together with the NU-WRF meteorological wind information, to
estimate urban $NO_x$ emissions. In other words, here we assume perfect knowledge of the winds and do not further consider the
impact of errors in $w$. As in previous studies, we only analyze data from April to September, in order to exclude winter data that
have larger uncertainties and longer $NO_x$ lifetimes. We also investigate the impact of the inclusion of winter data in Sect. 3.3.1; it
is found to be associated with a larger uncertainty. We further compute total emissions for each city, $Emis,$ by summing $E(x)$.

We perform a nonlinear least-squares fit of $f(x)$ to the observed line densities under windy conditions, $LD_{windy}(x)$, with $\tau$ as the
single fitting parameter. We use the package of scipy.optimize.curve_fit from the Python software library to perform the fitting.
We set the fit interval to 150 km in downwind direction, which corresponds to the e-folding distance for $\tau = 6$ h and $w = 7$ m s$^{-1}$.
The fit interval in the upwind direction and the $y$ direction are set to half the e-folding distance (75 km); the resulting area is large
enough to cover a highly populated and spread-out metropolitan region such as New York City. The definition of the fit interval
in upwind and downwind direction, and the across-wind integration interval are illustrated in Fig. 3. Note that we use $LD_{calm}(x)$
over a larger horizontal interval of 450 km to calculate the convolution in Eq. (2), in order to eliminate the edge effect of
convolution. Fitting results of insufficient quality (i.e., the correlation coefficient $R$ between the fitted and observed $NO_2$ LD < 0.9,
and one standard deviation error of $\tau > 10\%$) are discarded. We infer emissions simultaneously by summing $E(x)$ in Eq. (1). We
perform the fit for all wind direction sectors and then average the fitted $\tau$ and corresponding total emission $Emis$ with good quality,
using the fit residuals as inverse weights, to yield a best estimate of $<\tau>$ and $<Emis>$ for a given city. The standard deviation of the
fit results for different wind directions has been used to quantify uncertainties in Sect. 3.3.2.

We develop the new model function aiming for determining emissions for mix-sources, instead of isolated sources within a clean
background considered by Beirle et al. (2011). It is also different from that of Liu et al. (2016b), which was developed for complex
sources, but adopted an additional model function to fit emissions in a separate step. More comparisons with those two previous
methods will be discussed in Sect. 3.2.

We use the city of New York as a case study to demonstrate our approach. This city is well suited for illustrating the strength of
MISATEAM to estimate emissions for mixed sources, because it is a large city with multiple point and areal sources and is
surrounded by many other large sources. Figure 3 displays the complex spatial distribution of sources around New York. Under





southwesterly wind, the city of Philadelphia is located in the upwind direction and Long Island is located in the downwind direction, both of which are significant $NO_x$ sources.

We use wind fields averaged from the surface to 1000 m altitude for $w$ in this study. The synthetic $NO_2$ VCDs around New York are sorted by wind directions (Figure S1). Figure 4A displays the observed line densities for calm (blue circles) and southeasterly
winds (red circles) around New York and the fitted model function $f(x)$ (red lines). Generally, $f(x)$ describes the observed downwind patterns very well; the coefficients of determination ($R^2$) between observation and fit are 0.90–0.98 for different wind directions. Results for three wind direction sectors are discarded due to the fitting results being of insufficient quality. The resulting lifetimes show a range of 2.2–2.9 h, which result in emissions of 754–996 mol h$^{-1}$ for different wind directions, as shown in Fig. 4A–E.

### 2.3 Impact of temporal variations in wind fields

CTM-independent emission estimation algorithms usually assume a steady wind field over the duration of $NO_x$ lifetime. In the demonstration in Sect. 2.2, we use the wind fields sampled at the satellite overpass time to drive MISATEAM, consistent with previous studies (e.g., Beirle et al., 2011; Valin et al., 2013; Lu et al., 2015; Liu et al., 2017, 2020; Goldberg et al., 2019). This is expected to be reasonable for species with a short lifetime of a few hours such as $NO_x$ near noon of non-winter seasons. In reality, wind fields are variable over the $NO_x$ lifetime. Consequently, $NO_x$ emitted at a time prior to the satellite overpass may be
transported under different wind conditions than those at the overpass time.

Figure 5 illustrates the temporal variations in wind fields around New York. We use the southwesterly wind direction (with a valid fitting result) for demonstration. We select southwesterly winds observed at the overpass time as the baseline and find their backward trajectories for up to 8 hours. The backward trajectories are given at a time step of one hour. Not surprisingly, winds are not constant during the 9 hours from 8 hours before the overpass time to the exact hour of the overpass time. However, their
temporal variations are very small; the percentage changes of wind speeds are less than 5 % on average. Such minor changes have been confirmed for other wind directions with valid fitting results as well. For wind directions without good fit results, we observe larger variations. For instance, the percentage change of projected wind speeds among the same 9 hours reaches up to 34 % for the easterly wind (Fig. S2). These results shed light on the robustness of MISATEAM's steady-wind assumption. It is most likely that the fit fails when the assumption of steady wind is not satisfied. In other words, the inherent fitting assumption is robust when the
fit results have sufficient quality as defined in Sect. 2.2.

We perform sensitivity analyses to investigate the potential impact of temporal variations in winds on the fit results. We extend the time windows used for calculating averaged wind fields from 1 h (i.e., at the overpass time) to 3 h (i.e., starting from the overpass time and extending into the past 2 h), 6, 9, and 12 h. We weight the winds based on their temporal proximity, i.e., the wind closer to the overpass time is given larger weight, following Eq. (3).

$$\overline{w_{i,d}} = \frac{\sum_{h=0}^{N} w_{h,i,d} \times e^{-h/t_0}}{\sum_{h=0}^{N} e^{-h/t_0}},$$  (3)

Where $h$ represents the number of hours prior to the overpass time. $i$ and $d$ denote an individual grid cell and day, respectively. $w_{h,i,d}$ is the wind for a specific grid cell $i$ on day $d$ at the time of $h$ hours prior to the overpass time. $N$ is the length of the time window used (units of hour). The weighted average winds $\underline{w_{i,d}}$ are further applied with MISATEAM to infer $NO_x$ lifetimes and emissions for investigated cities. We set $t_0$ to a constant value of 3 derived from rounding the average NU-WRF lifetimes for all
investigated cities (see details in Sect. 2.4). The fitting results are found to be relatively insensitive to the choice of $t_0$. The differences of the fitted lifetimes and emissions are -2 ± 15 % and 3 ± 16 %, respectively, when we increase $t_0$ by a factor of 2.


This is significantly smaller than the difference between the fit results based on weighted average winds and the winds at the overpass time (shown in Sect. 3.1).

### 2.4 Performance evaluation

In order to evaluate the fitting results, we infer given emissions and NU-WRF lifetimes from the NU-WRF inputs/outputs. The given emission *Emis'* is derived by summing up NU-WRF $NO_x$ emissions from all grid cells within the fit interval. The NU-WRF lifetime *τ'* can be computed by solving Eq. (1), i.e.,

$$\tau' = \frac{\sum ratio \times [LD_{calm}(x) - b]}{\sum E'(x)},$$

(4)

where *E'(x)* is the given $NO_x$ emission line densities under calm wind conditions, as function of distance *x* from the city center.

For evaluation, we compute the correlation coefficient (*R*), the Normalized Mean Bias (NMB), and the Root Mean Squared Error (RMSE) of the fitted emissions and the given emissions for all investigated cities. The model performance metrics of NMB and RMSE for the emission (*Emis*) evaluation are defined as

$$NMB = \frac{\sum_{i=1}^{n} (Emis_i - Emis_i')}{\sum_{i=1}^{n} Emis_i'},$$

(5)

and

$$RMSE = \sqrt{\frac{\sum_{i=1}^{n} (Emis_i - Emis_i')^2}{n}},$$

(6)

respectively, where *i* represents the individual city and *n* is the total number of cities used for evaluation. The metrics for lifetime evaluation are consistent with Eqs. (5) – (6) when replacing *Emis* with *τ* and *Emis'* with *τ'*. A good method should have a large *R*, a near-zero NMB, and a small RMSE.

### 3 Results and Discussion

### 3.1 Evaluation

We apply MISATEAM to 60 large cities over the US (see the selection criteria of cities in Sect. 2.1). For 5 cities, we are not able to initiate the fitting procedure, due to lack of observations under calm wind conditions to calculate $LD_{calm}(x)$. We derive valid fitting results for 26 cities. The locations of the 26 cities are shown in Fig. 2. The other 29 cities without valid results either had small correlation coefficients (< 0.9) or large fitting errors (standard deviation error of *τ* > 10 %); those cities tend to have larger

temporal variations in winds (similar to Fig. S2), which do not satisfy MISATEAM's requirement for steady winds prior to satellite overpass.

Figure 6 compares MISATEAM estimated lifetimes and emissions with the NU-WRF lifetimes and given emissions for the 26 cities. The comparison shows good consistency in general. For results derived from the wind data sampled at the overpass time (hereafter referred to as "1 h wind"; red dots), values of *R* are 0.56 and 0.88 for lifetimes and emissions, respectively. The bias is

rather small for the lifetime comparison with NMB of -0.04 and RMSE of 0.54. The bias is larger for emissions, primarily caused by the assumption of a constant $NO_x$ to $NO_2$ ratio (*ratio*). The errors arising from the differences between *ratio* for individual cities and a constant value of 1.32 will be propagated into the resulting emissions. The impact of the prescribed *ratio* on inferring emissions will be discussed in more detail in this section (Fig. 7).





The use of wind data over 9 hours prior to the overpass time improves the performance of MISATEAM. Figure 6 compares the
inferred lifetimes and emissions based on the 9 h weighted average of wind data (hereafter referred to as "9 h wind"; blue dots).
The results derived from the weighted average wind data show larger correlations with $R$ increasing from 0.56 to 0.79 for lifetimes
and from 0.88 to 0.96 for emissions; and smaller bias with NMB decreasing from -0.04 to 0.02 for lifetimes and from 0.23 to 0.13
for emissions, when comparing with those derived from the 1 h wind. We have performed the comparison using results based on
the weighted averages of 3 h, 6 h, and 12 h wind data as well. The usage of wind information prior to the satellite overpass time
succeeds in increasing the agreement in all these cases (Fig. S3).

The importance of applying wind information prior to the satellite overpass time should not be overinterpreted. The fitting function
Eq. (2) by definition is not capable of describing the $NO_2$ plumes under significantly varying wind directions because such temporal
variations are not considered in the fitting function. In this way, wind directions and the results inferring from different wind
scenarios are not expected to vary significantly, as far as fits with sufficient quality are yielded. Only 6 out of 26 cities show
relative differences larger than 20 % when comparing results derived from 1 h wind to those derived from 9 h wind.

We examine a scenario, namely "NU-WRF $NO_x/NO_2$", to investigate possible errors from the assumption of a constant *ratio*. We
replace *ratio* with the ratio of $NO_x$ and $NO_2$ calculated directly from $NO_2$ and NO VCDs per grid cell by NU-WRF outputs, and
then use MISATEAM for inferring $NO_x$ emissions. Any difference in the inferred emissions compared to the emissions based on
the prescribed ratio of 1.32 ("this study") can then be assumed to originate from errors in the assumption of *ratio*. Figure 7 compares
the results using a prescribed ratio (blue dots) with those using NU-WRF $NO_x/NO_2$ (grey dots). The comparison shows nearly the
same correlations to the given emissions, but a smaller bias for results based on NU-WRF $NO_x/NO_2$ with NMB dropping from
0.13 to almost zero (0.03). The comparison suggests that the influence of changing ratio on derived emissions is limited, because
its spatial variation is significantly smaller than that of $NO_x$ lifetime and $NO_2$ columns ($\tau$ and $LD_{calm}(x)$ in Eq. (1)). Considering the
investigated cities are located all over the country and have a wide range of geographic features, we conclude that a constant ratio
is a reasonable assumption without resulting in significant bias to the derived emissions for typical US cities. The errors associated
with the assumption is estimated to be 10 %, consistent with our previous estimates based on literature reviews (Beirle et al., 2011;
Liu et al., 2016b). However, for applications based on geostationary satellites with changing local observation time, the approach
using a constant value for *ratio* is subject to larger uncertainties arising from the diurnal cycle of *ratio* (Han et al., 2011).

We examine an additional scenario, namely "constant lifetime", to show the necessity of deriving lifetimes for individual cities.
Instead of individually fitted lifetimes for each city, we use the mean NU-WWRF lifetime of all cities (2.5 h) for the calculation
of emissions in the "constant lifetime" scenario. The emissions correlation drops to -0.03 (Fig. 7), showing that individually fitted
lifetimes are critical for this method. The bias is also larger with RMSE increasing by a factor of 3 compared to results based on
the individually fitted lifetimes. This further improves our confidence that the derived variation of the fitted lifetimes carries
important information on local variability of the oxidizing capacity of urban plumes. The individual lifetimes are well suited for
the determination of emissions, suggested by the significantly improved consistency with given emissions.

### 3.2 Comparison with previous methods

We further evaluate MISATEAM by comparing the results with those derived from two previous approaches including Beirle et
al. (2011) and Liu et al. (2016b). We apply all three approaches to fit lifetimes and emissions for all 26 cities investigated by this
study. Note that we use the 9 h wind for all approaches for best performance and consistency.

Figure 8A illustrates the comparison for inferring lifetimes. The approach of Beirle et al. (2011) does not predict lifetimes well,
with a poor correlation ($R = 0.01$). This is not surprising, because by definition the method can only represent a single point source





convolved with a Gaussian function, and was not intended to be applied to mixed-sources with interfering emissions from nearby cities or industrial areas. For instance, it is capable of giving an accurate estimate for an isolated city of St. Louis in Missouri, with a relative difference of less than 10 % compared to the NU-WRF lifetime. However, for most cities with a polluted background, the fitted lifetimes are biased significantly due to the interference from surroundings. This is consistent with the previous findings for this approach: an additional source at 100 km with only 10% of the emissions of the source under investigation causes a lifetime bias of 20 %; for an interfering source of the same order as the source of interest, the method fails completely (Liu et al., 2016b). Several studies adopted a plume rotation technique (Pommier et al., 2013; Valin et al., 2013) to advance the approach of Beirle et al. (2011), which is not applicable to mixed-sources as well. These techniques rotate $NO_2$ measurements centered over the city center so that $NO_2$ columns under different wind directions are aligned in a common upwind-to-downwind direction. This increases the number of observations used for analysis without introducing additional errors for (quasi) point sources, compared with individually analyzing observations by wind directions as done in this study. However, for mixed-sources investigated in this study, use of such rotation techniques may result in significant bias by allocating the $NO_2$ from interfering sources into a ring of elevated $NO_2$ values around the source of the interest and thus amplifying the $NO_2$ signal of the source. An illustration of this amplification can be found in Fig. S2 of Fioletov et al. (2015).

It is interesting to note that the performance of the approach of Liu et al. (2016b) is also worse than MISATEAM, although they share the same concept of using the $NO_2$ patterns observed under calm wind conditions as proxy of emission patterns instead of assuming a single point source as in Beirle et al. (2011). It is most likely that Liu et al. (2016b) overfits the model by introducing too many degrees of freedom. As suggested by Fig. 8A, the model function of Liu et al. (2016b) occasionally tries to "explain" changes of scaling factors by a shorter lifetime, resulting in a small $R$ of 0.21. In MISATEAM, we decrease the number of fitting parameters from three in Liu et al. (2016b) to only one (see details in Eq. (A4) of Appendix A), which improves the robustness of the fit results.

Emission comparisons in Fig. 8B show better agreement than lifetime comparisons in Fig. 8A for all approaches. MISATEAM-derived emissions show the best consistency with the given emissions. According to mass balance, the magnitude of emissions equals the total mass of $NO_x$ divided by lifetime. In the evaluation for MISATEAM, the given emissions range from 57 mol h$^{-1}$ to 717 mol h$^{-1}$ for all investigated cities, the variation in which is significantly larger than that in lifetimes ranging from 1.5 h to 3.7 h. This finding also holds for the other two approaches. The other two approaches can achieve a good correlation with the given emissions by providing reasonable estimates for the magnitude of the total $NO_x$ mass, even though they fail to predict variations in lifetimes between cities. For instance, the results derived from the approach of Liu et al. (2016b) show a small $R$ of 0.21 to the NU-WRF lifetimes, but a significantly stronger correlation to the given emissions of 0.94, which is comparable to that of MISATEAM-derived emissions. But Liu et al. (2016b)-derived emissions are still associated with larger biases arising from the estimates for lifetimes. The values of NMB are 0.13 and -0.21 for emissions derived from MISATEAM and the approach of Liu et al. (2016b), respectively, when comparing against the given emissions. Note that the derived and given emissions from the approach of Liu et al. (2016b) is smaller than the two other approaches, but does not indicate a smaller bias. Liu et al. (2016b) only aims to estimate emissions from the city center, considered as a (quasi) point source, instead of all sources in the urban area. In this way, *emis* and *emis'*, and thus RMSE are smaller than that for MISATEAM.

### 3.3 Uncertainty analysis

The good consistency in Sect. 3.1 increases our confidence that the fitted lifetimes and emissions represent the real-word characteristics well. We investigate their uncertainties in this session.




### 3.3.1 Sensitivity analysis

Analogous to Beirle et al. (2011) and Liu et al. (2016b), we investigate the impact of the a-priori choice of fit and integration intervals, and wind layer height. The dependency of the fit results for $\tau$ and *Emis* on these three choices is tabulated in Table 1.

The fitted lifetimes are generally robust with respect to changes of the fit and integration intervals, because the $NO_2$ distribution under calm wind conditions, $LD_{calm}(x)$, is a good representation of the emission pattern in any case. An increase of the fit interval in downwind direction by 50 km affects the resulting lifetimes by about $-2 \pm 11$ %. The changes of derived lifetimes are also small when increasing the fit interval in upwind direction ($4 \pm 12$ %) or integration interval (interval $i + 25$ km in Fig. 3; $4 \pm 18$ %) by 50 km. MISATEAM succeeds in avoiding choosing intervals city by city manually, which has been done in previous studies in order to minimize the influence of other nearby sources.

We use the change of the ratio of fitted emissions *emis* to given emissions *emis'*, $\Delta(emis/emis')$, to show the impact of the enlarged fit and integration intervals on emissions. We do not focus on the change of emissions, $\Delta emis/emis$, because the fitted emissions are expected to be sensitive to the enlarged intervals which include additional sources and thus more emissions. The fitted emissions show an average growth of 48 % associated with extending the integration interval by 50 km to increase the given emissions by 36 %. The rise in emissions is similar to increasing the fit interval in the upwind direction by 50 km with 33 % for the given emissions and 42 % for the fitted emissions. However, for the scenario of a larger downwind-direction, upwind-direction and integration interval, the change of the ratio of *emis* to *emis'* is rather small, which is 3, 9, and 9 % on average, respectively; the fitted emissions show good consistency with the given emissions, reporting the correlation coefficient of 0.95, 0.92, and 0.86, respectively. It is interesting to note that the fitted emissions are rather insensitive to the extension of the fit interval in downwind direction. Neither the given nor the fitted emissions are significantly changed by increasing the downwind-direction interval from 150 km to 200 km. It suggests that we succeed in capturing the complete downwind plume and reaching the background areas by the default setting of 150 km for the investigated cities in this study.

Uncertainties associated with the choice of layer height (e.g., 500 m, 1000 m, or 2000 m) are relatively small. The resulting lifetimes and emissions change about 16 % and -9 % on average when averaging the wind fields from surface to up to 500 m. The average changes are -11 % and 13 % for inferring lifetimes and emissions, respectively, when adopting the wind layer height of 2000 m. This is consistent with the findings in the previous studies (e.g., Beirle et al., 2011 and Liu et al., 2016b).

We also apply MISATEAM to year-round $NO_2$ data to investigate the impact of including winter data on the performance of the method. We keep default settings of MISATEAM as described in Sect. 2.2 for the fit. As expected, the fitted results differ more significantly from given values compared with results based on using only non-winter data. The bias is larger with NMB changing from 0.02 to -0.14 for lifetimes and from 0.13 to 0.27 for emissions. This improves our confidence that MISATEAM, most likely its inherent steady-wind assumption, is more vulnerable during the winter season with longer $NO_x$ lifetimes.

### 3.3.2 Uncertainty quantification

We calculate the uncertainties of inferred results based on the fitting metrics (Fig. 6 and 7) and the dependencies on the a priori settings as investigated in the above sensitivity studies. We attribute uncertainties of 15 % and 20 % to the derived lifetimes and emissions, respectively, based on the mean of relative differences for all 26 cities (14 % for lifetimes and 21 % for emissions). The derived emissions have higher uncertainties arising from uncertainty in the $NO_x$ to $NO_2$ scaling factor. The derived emissions in terms of $NO_2$ are upscaled to $NO_x$ based on a constant $NO_x/NO_2$ ratio of 1.32, representing typical urban conditions at noon (Seinfeld and Pandis, 2006). Since MISATEAM aims to provide estimates for cloud-free satellite observations at the overpass time





close to noon of non-winter seasons and it focuses on polluted regions with generally high tropospheric ozone, this value is reasonably accurate. However, the $NO_x/NO_2$ ratio might vary locally. NU-WRF reports $1.4 \pm 0.1$ with a range of $1.2 – 1.6$. The overall impact of variations in this ratio is shown to be relatively small (see Sect. 3.1).

We can identify additional uncertainties that would be present when applying MISATEAM to "real" data instead of synthetic data. The uncertainty of satellite $NO_2$ observations propagates into the uncertainty of emissions. The uncertainty of satellite $NO_2$ observations has less impact on the lifetime estimation and only results in errors for lifetimes when satellite observations have systematic errors depending on the distance from the source. The total uncertainty of $NO_2$ VCDs results from uncertainties in the spectral fit in the retrieval, the stratospheric and tropospheric separation, and the tropospheric air mass factor (AMF). In the model

function of MISATEAM, a possible bias associated with the stratospheric and tropospheric separation is eliminated by use of the background term $b$. The uncertainty in the spectral fit in the retrieval is rather small compared to that associated with AMF (Boersma et al., 2007). We estimated the overall uncertainty primarily arising from the uncertainty in the tropospheric AMF is about 25 % based on validation of TROPOMI $NO_2$ products with ground-based measurements (e.g., Griffin et al., 2019; Ialongo et al., 2020; Zhao et al., 2020). Since the random uncertainty of the tropospheric $NO_2$ observations could be suppressed due to the

consideration of long-term means, this estimate may be conservative.

The presence of clouds is an additional source of uncertainties. We are required to exclude satellite observations with significant cloud fractions in the instrument's field of view. For TROPOMI $NO_2$ products, we usually remove data with cloud radiance fraction $\geq 0.5$. A bias is observed in $NO_2$ VCD averages as a result of removing the data during cloudy conditions (Geddes et al., 2012). The bias is associated with changing photochemistry, meteorology, and pollutant transport, which may also have impacts on

$NO_x/NO_2$ ratio and $NO_x$ lifetime. The magnitude of a bias is expected to vary from city to city. We calculate the fraction of cloudy scenes to total scenes over the fit domain of individual cities based on TROPOMI $NO_2$ products from April through September, 2020. The fractions range from $16 – 56$ % for the considered cities. For cities with heavy cloud cover, like New York, Philadelphia, and Washington D.C. with a fraction > 50%, the impact associated with cloud selection criteria is expected to be larger than cities with more clear sky. We estimated an uncertainty of 10% arising from cloud selection criteria based on the evaluation performed

at urban sites (Geddes et al., 2012).

Additionally, the accuracy of wind fields contributes to the uncertainties of both lifetimes and emissions. It can affect the sorting of the $NO_2$ VCDs according to wind directions as well as the conversion of the downwind decay from a function of distance into a function of time in Eq. (2). We estimate the uncertainties associated with the wind data to be approximately 30 % based on the comparison of wind information between reanalysis product and sounding measurements (see Table S3 in Liu et al., 2016b).

We define total uncertainties of the resulting lifetimes and emissions as the root of the quadratic sum of the above-mentioned error contributions that are assumed to be independent. We estimated that total uncertainties of $NO_x$ lifetime and emissions for a US city are 43 % and 45 %, respectively.

## 4 Conclusions and future work

In this work we developed a CTM-independent approach, MISATEAM, to infer $NO_x$ lifetimes and emissions from satellite $NO_2$

observations. As in Liu et al. (2016b), MISATEAM is developed for sources with polluted backgrounds. It adopts the approach of using $NO_2$ spatial patterns under calm wind conditions as a proxy of the spatial patterns of emission sources to account for interferences from neighboring strong sources. MISATEAM improves upon Liu et al. (2016b) by advancing the fitting function to reduce the number of parameters and to provide estimations of $NO_x$ lifetimes and emissions simultaneously.



We applied MISATEAM to synthetic tropospheric $NO_2$ VCDs over the continental US provided by a NU-WRF high resolution
model simulation. We found that our new method for determining $NO_x$ lifetimes and emissions was applicable to 26 cities. The
derived results were generally in good agreement with the NU-WRF given values. In existing studies, wind fields sampled
simultaneously with satellite observations were used to drive the CTM-independent approach. We investigated the impact of
temporal variations in winds on fitted results and found the use of wind data prior to satellite overpass time improves performance
of our approach. $R$ between inferred and NU-WRF lifetimes increased from 0.56 to 0.79 and for emissions increased from 0.88 to
0.96 when comparing results based on 1 h and 9 h winds, respectively. The comparison between MISATEAM and the approaches
of Beirle et al. (2011) and Liu et al. (2016b) suggests that MISATEAM is more suitable for non-isolated sources, particularly for
lifetime estimation. Lifetimes inferred from the previous approaches showed rather weak correlations with respect to NU-WRF
lifetimes (0.01 for Beirle et al. (2011) and 0.21 for Liu et al. (2016b)) as compared with that from MISATEAM (0.79).

We plan to apply MISATEAM to observations from TROPOMI and geostationary satellite instruments including the Korean
Geostationary Environmental Monitoring Spectrometer (GEMS; Kim et al., 2012), NASA Tropospheric Emissions: Monitoring of
Pollution (TEMPO; Chance et al., 2012), and ESA Sentinel-4 (Ingmann et al., 2012). These instruments have spatial resolutions
similar to the NU-WRF simulation (4 km) used in this study. We estimate that uncertainties in $NO_x$ lifetime and emissions arising
from MISATEAM are approximately 15% and 20%, respectively, for typical (US) cities. Additional uncertainties are associated
with wind errors in the reanalysis dataset as well as errors in the satellite $NO_2$ retrievals. We will attempt to reconcile bottom-up
and satellite-derived urban emissions to generate a merged inventory (e.g., Liu et al., 2018) to provide timely $NO_x$ emissions
estimation for air quality and climate modeling communities.

*Data availability.* The NU-WRF model outputs are available upon request from Zhining Tao (zhining.tao@nasa.gov). Additional
data related to this paper may be requested from the corresponding author.

*Author contributions.* Conceptualization and methodology: F.L., J.J., and S.B.; Model simulation: Z.T.; Formal analysis: F.L. and
Y.Y.; Writing—original draft: F.L.; Writing—review and editing: All authors; Visualization: F.L.; Supervision, project
administration, funding acquisition: F.L. and J.J.

*Competing interests.* The authors declare that they have no competing interests.

*Acknowledgements.* This work was funded by NASA through the Aura project data analysis program and through the Atmospheric
Composition Modeling and Analysis Program (ACMAP) program (grant no. 80NSSC19K0980).

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

**Appendices**

Appendix A Derivation of the model function *f(x)*

We derive Eq. (1) based on the continuity equation for steady state, following Eqs. (A1) – (A2) given by

$$E(x) = S(x) + D(x),\tag{A1}$$





$$S(x) = \frac{ratio \times LD(x)}{\tau},$$ (A2)

where $E(s)$, $S(x)$ and $D(x)$ represent the line densities of $NO_x$ emission, sink and divergence, respectively. As the $NO_x$ sinks are

dominated by the chemical loss due to reaction of $NO_2$ with OH at the local overpass time of TROPOMI (13:30 local time), sink $S(x)$ can be described by a first order time constant $\tau$ and thus is proportional to the $NO_2$ line density $LD(x)$ itself as shown in Eq. (A2). Beirle et al. (2019) provided further details.

We use $NO_2$ line densities under calm wind conditions, $LD_{calm}(x)$, to simplify Eqs. (A1) – (A2). In principle, there is no $NO_x$ transport under perfect calm wind conditions (i.e., divergence $D(x)$ is zero), and thus the emission $E(x)$ equals the sink $S(x)$ given

by $\frac{ratio \times LD_{calm}(x)}{\tau}$. However, we use the threshold of 2 m s$^{-1}$, instead of 0 m s$^{-1}$, as the criterion for calm wind to get a good compromise between sufficient sample sizes for both the calculation of line densities for calm conditions as well as for windy conditions. In order to account for the error associated with this criterion and possible systemic differences between windy and calm wind conditions (e.g., cloud conditions, vertical profiles, or lifetimes), and to account for the upper tropospheric background column which is not driven by local emissions, we introduce a constant background $b$ in the fitting function, as given by Eq. (A3).

$$E(x) = S(x) = \frac{ratio \times [LD_{calm}(x) - b]}{\tau},$$ (A3)

We derive Eq. (2) following the concept proposed by Liu et al. (2016). We use $LD_{calm}(x)$ as a proxy for emissions instead of assuming a single point source as in previous studies (e.g., Beirle et al., 2011; Laughner et al., 2019). The $NO_2$ line density without considering the chemical decay is given by $\frac{E(x)}{ratio \times w}$ based on a Gaussian plume model. This formulation is different from the model function $f(x)$' originally proposed by Liu et al. (2016), which was given by

$$f(x)' = a \times LD_{calm}(x) * e^{-\frac{x}{w \times \tau}} + b,$$ (A4)

We replaced one fitting parameter, the scaling factor $a$ in $f(x)'$, with variables that have physical meanings in the new model function $f(x)$. The new formulation was shown to improve the model performance in Sect. 3.2. We then convolved $\frac{E(x)}{ratio \times w}$ with an exponential function $e^{-\frac{x}{w \times \tau}}$ describing the chemical decay to form the new model function $f(x))$, implicitly assuming a constant effective lifetime $\tau$.






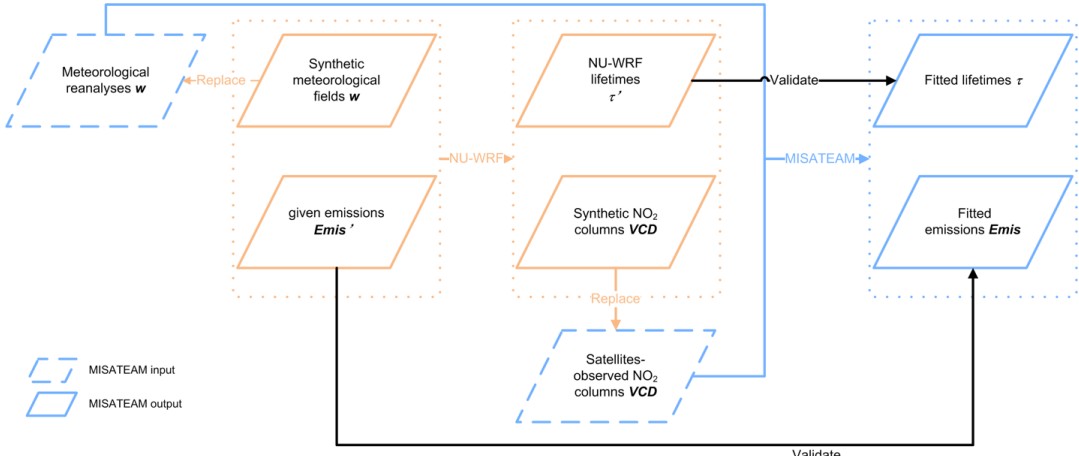

**Figure 1 Schematic of our evaluation system to assess the accuracy of the inferring NO$_x$ lifetimes and emissions derived from MISATEAM. The blue symbols represent the inputs and outputs of MISATEAM. The orange symbols represent the information derived from NU-WRF.**


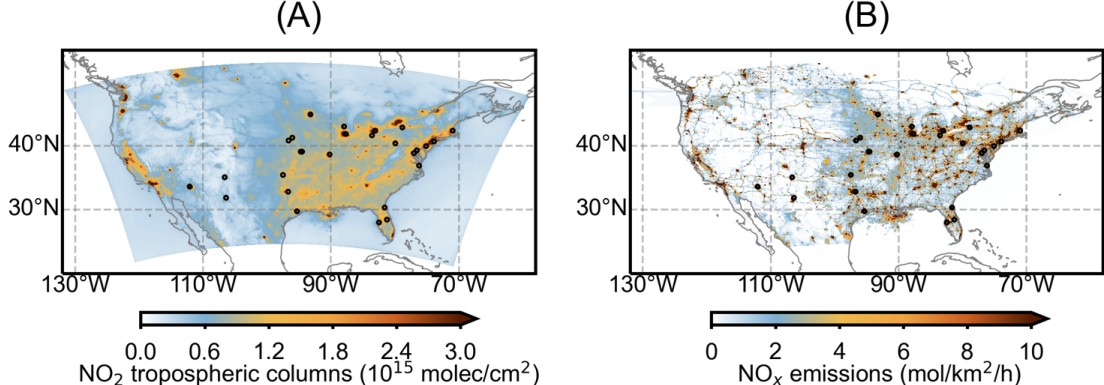

**Figure 2 Domain used for simulation. (A) Mean NU-WRF tropospheric NO$_2$ vertical column densities. (B) Mean NEI NO$_x$ emissions fluxes used to drive the NU-WRF simulation. Hourly mean data at the local overpass time of TROPOMI are averaged from April through September, 2016. Locations of the 26 cities investigated in this study are labelled by circles (see Section 3).**


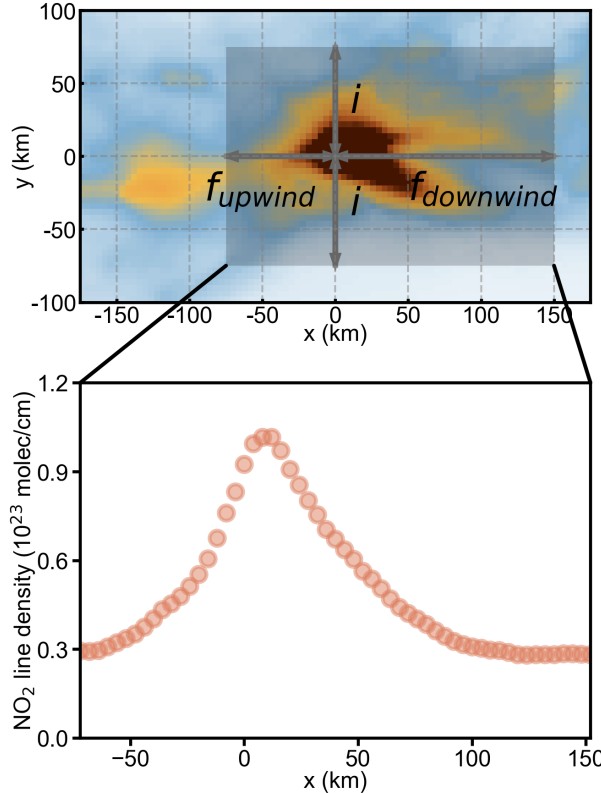

**Figure 3 Sketch of the definition of Line Densities. For each wind direction, mean NO₂ VCDs are integrated in across-wind direction y over the interval *i*, resulting in line densities *LD(x)*. The fit is performed over the entire upwind interval (*f_upwind*) and downwind intervals (*f_downwind*). The city center is the coordinate origin. The top panel shows the NU-WRF tropospheric NO₂ VCDs around New York City under southwesterly wind, however the image is rotated by 45 degrees in the clockwise direction to present NO₂ VCDs in an upwind-downwind direction. The city of Philadelphia and Long Island are located in the upwind and downwind direction, respectively.**




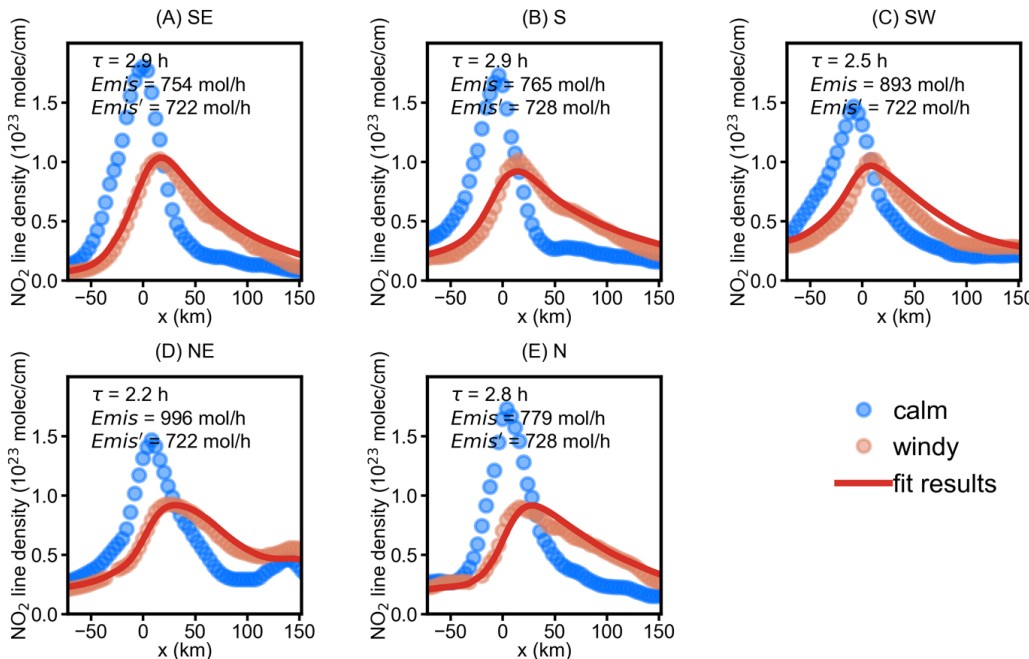

**Figure 4 NO₂ line densities around New York for different wind direction sectors. Circles: NO₂ line densities for calm (blue circles) and (A) southeasterly, (B) southerly, (C) southwesterly, (D) northeasterly, and (E) northerly winds (red circles) as a function of the distance *x* to New York center. Red line: the fit result *f(x)*. The numbers indicate the fitted NOₓ lifetime (*τ*), derived emissions (*Emis*) and given emissions (*Emis'*). NO₂ line densities are derived from NO₂ VCDs averaged from April through September, 2016. NO₂ line densities for the remaining wind direction sectors are discarded due to the fitting results being of insufficient quality.**


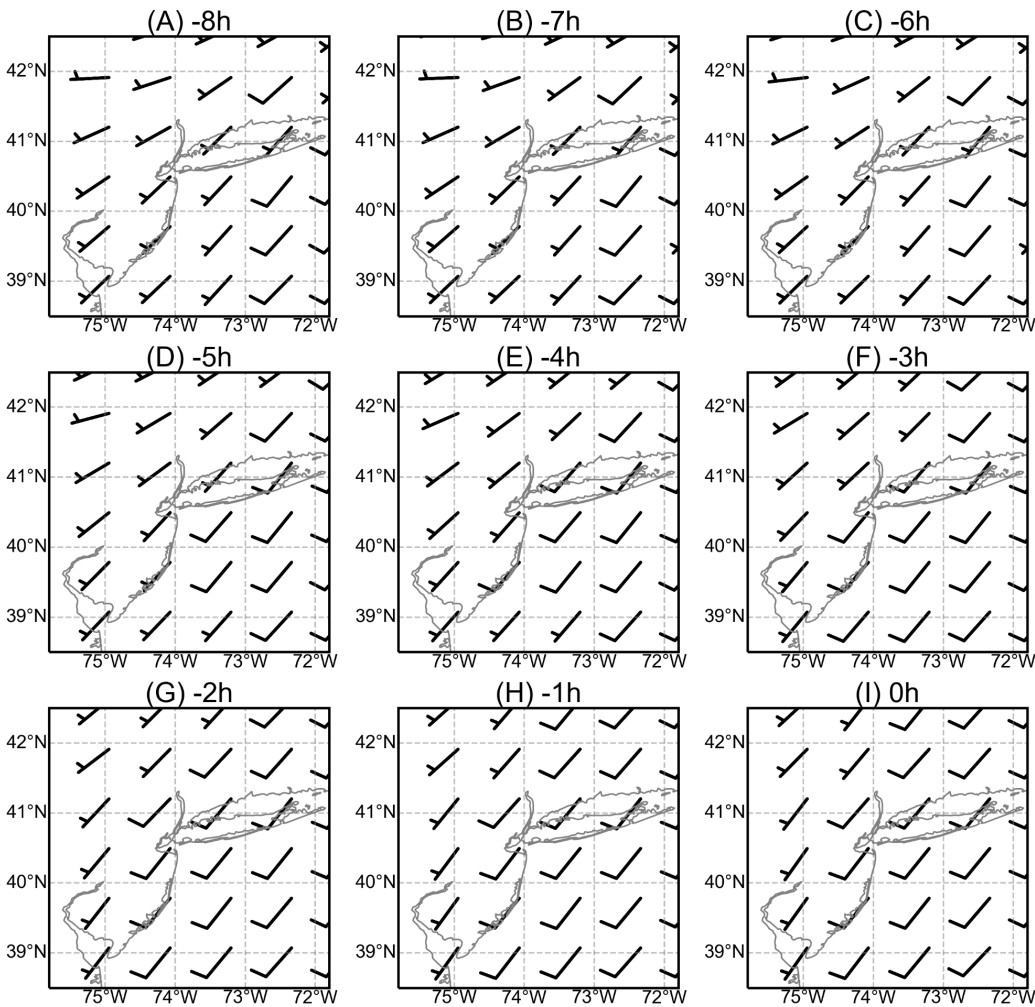

**Figure 5 Wind barbs around New York City for different times of the day. All southwesterly winds at the local overpass time of TROPOMI from April to September of 2016 are averaged and shown in (I). Wind barbs for the southwesterly winds backward trajectories from 8 to 1 h prior to the overpass time are displayed in (A) – (H). Wind speed is given in the units of knots, which is a nautical miles per hour (1.9 km per hour). Each short and long barb represents 5 knots (9.3 km/h) and 10 knots (18.5 km/h), respectively.**





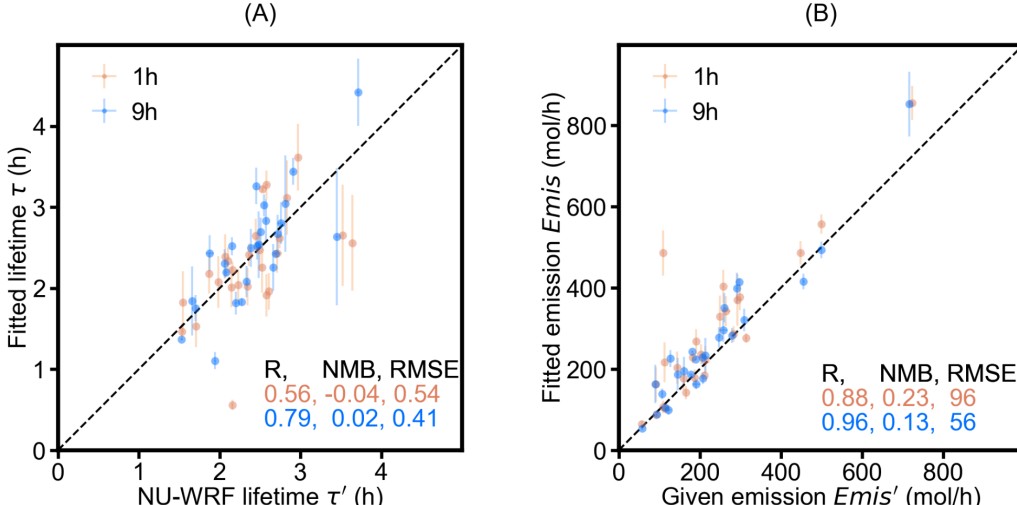

**Figure 6 Scatterplots of (A) the fitted NO$_x$ lifetime $\tau$ as compared to the NU-WRF lifetime $\tau'$; and (B) the fitted NO$_x$ emissions *Emis* as compared to the given emissions *Emis'*. Error bars show the standard error of the fitted results for all available wind directions.**

**Standard error is defined as standard deviation divided by $\sqrt{n}$, with $n$ being the number of available wind directions. The results deriving from the wind fields sampled at the TROPOMI overpass time ("1 h") and the weighted average of 9 h wind fields ("9 h") are displayed by red and blue dots, respectively. The dash line represents the 1:1 line.**

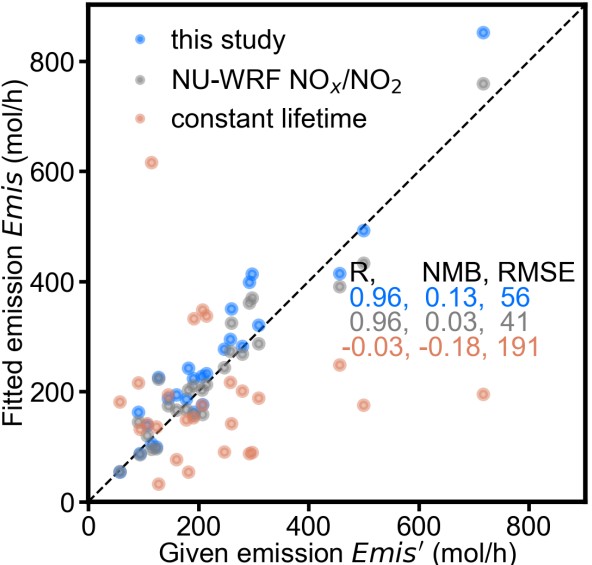


**Figure 7 Scatterplot of the fitted total NO$_x$ emissions *Emis* as compared to the given total emissions *Emis'* under different scenarios. The blue, grey and red dots represent the scenarios based on the fitted lifetime $\tau$ and a constant NO$_x$ to NO$_2$ ratio of 1.32 ("this study"), the fitted lifetime $\tau$ and the NO$_x$ to NO$_2$ ratio given by NU-WRF model ("NU-WRF NO$_x$/NO$_2$"), and a constant lifetime of 2.5 hours and a constant NO$_x$ to NO$_2$ ratio of 1.32 ("constant lifetime"), respectively. The dash line represents the 1:1 line. Statistics provided in the**

**inset table.**

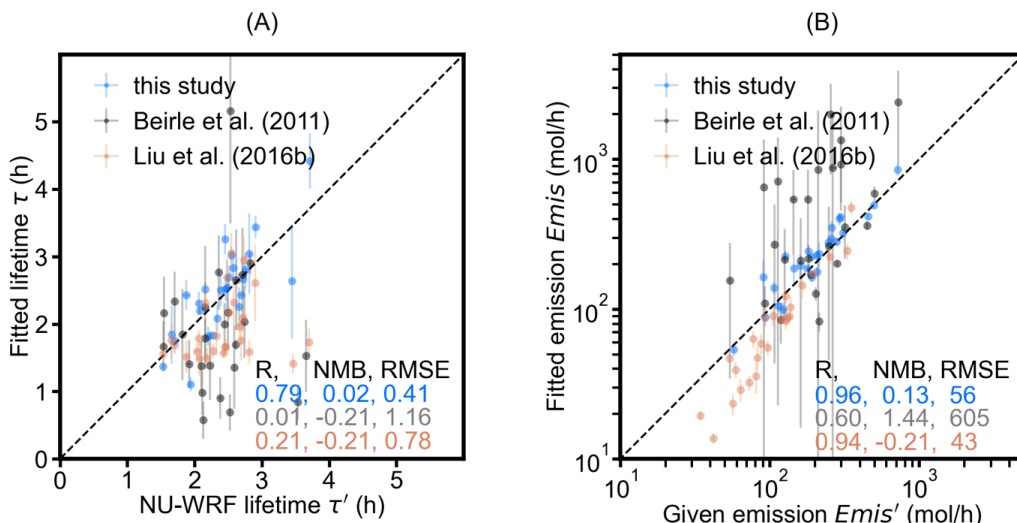

**Figure 8 Scatterplots of (A) the fitted NOₓ lifetime $\tau$ as compared to the NU-WRF lifetime $\tau'$; and (B) the fitted NOₓ emissions *Emis* as compared to the given emissions *Emis'*. Error bars show the standard error of the fitted results for all available wind directions. Standard error is defined as standard deviation divided by $\sqrt{n}$, with *n* being the number of available wind directions. The results derived from MISATEAM, the approach of Beirle et al. (2011), and the approach of Liu et al. (2016b) are displayed by blue, grey and red dots, respectively. The dash line represents the 1:1 line. Note that figure B is plotted in a logarithmic scale.**

Table 1. The mean relative change of lifetimes and emissions for different choices of fit and integration intervals, and wind fields.

| | Interval$_{downwind}$[a] + 50 km | Interval$_{upwind}$[a] + 50 km | Interval$_{integrate}$[a] + 50 km | 500 m[b] | 2000 m[b] |
|---|---|---|---|---|---|
| mean[$\Delta\tau'/\tau'$] | 0% | 9% | 8% | 2% | -2% |
| mean[$\Delta\tau/\tau$] | -2% | 4% | 4% | 16% | -11% |
| mean[$\Delta emis'/emis'$] | 0% | 33% | 36% | 0% | 0% |
| mean[$\Delta emis/emis$] | 2% | 42% | 48% | -9% | 13% |
| mean[$\Delta(emis/emis')$] | 3% | 9% | 9% | -8% | 12% |

[a]Interval$_{downwind}$ = 150 km, Interval$_{upwind}$ = 75 km, Interval$_{integrate}$ = 150 km

[b]the wind fields are averaged from the surface up to this height