# Peer review of "A new method for inferring city emissions and lifetimes of nitrogen oxides from high-resolution nitrogen dioxide observations: A model study"

_Atmospheric Chemistry and Physics, 2021_

## Author Response (AR1)

Below we reply to the reviewer comments point by point. The reviewer comments are shown in *italic*, and corresponding modifications and citations of the manuscript are quoted.

Anonymous Referee #1

*This manuscript describes an enhanced method to infer NOx emissions from urban areas using satellite data. This manuscript is a proof of concept and validation study tested on 60 US cities. This manuscript is a nice advancement. I recommend minor revisions based on my comments below.*

**Response:** We thank Referee #1 for the encouraging comments. All comments and suggestions have been considered carefully and addressed below.

*The mention of "TROPOMI overpass time" initially added some confusion, since the mention of TROPOMI implied that TROPOMI data was somehow used. After re-reading, it is clear that TROPOMI data was not used and instead this method is applicable to any satellite with an early afternoon overpass time (e.g. OMI). Lines 120, 156 and figure captions of Figure 2, 5, and 6 should be revised to remove the word "TROPOMI" and instead use the actual model time, presumably 13:00 or 14:00 local time. Perhaps Line 120 should be revised to say ... "sampled at 13:00 local time, which approximately corresponds the early afternoon overpass time of OMI and TROPOMI", and then 13:00 (or whatever the exact time of the model output used) should be used in Line 167, and the figure captions.*

**Response:** We have made the revisions following the suggestion by updating "the TROPOMI overpass time" with "14:00 local time", as follows:

line 120: "Figure 2A illustrates the six-month average of the simulated hourly mean tropospheric $NO_2$ VCDs sampled at 14:00 local time, which approximately corresponds to the early afternoon overpass time of OMI and TROPOMI."

line 156: "we use the NU-WRF high-resolution tropospheric $NO_2$ VCDs sampled at 14:00 local time as the synthetic $NO_2$ VCD observations."

figure 2 caption: "Hourly mean data at 14:00 local time are averaged from April through September, 2016."

figure 5 caption: "All southwesterly winds at 14:00 local time from April to September of 2016 are averaged and shown in (I)."

figure 6 caption: "The results deriving from the wind fields sampled at 14:00 local time ("1 h") and the weighted average of 9 h wind fields ("9 h") are displayed by red and blue dots, respectively."

Figure S3 caption: "The results deriving from the wind fields sampled at the 14:00 local time ("1 h"), the weighted average of 3 h wind fields ("3 h"), 6 h wind fields ("6 h"), 9 h wind fields ("9 h"), and 12 h wind fields ("12 h") are displayed by red, yellow, green, blue, and grey dots, respectively."

*Section 2.1 could use a bit of reorganization. For example, "tau" is first discussed in Lines 144 - 147, and then other variables are mentioned and then "tau" is discussed again in Lines 162 - 173. Lines 144 - 147 should be discussion in succession with Lines 162 -173. This is also true of the "ratio" and "b" variables. They are first discussed in Line 137, and then again in Lines 148 - 154. This makes it hard to follow.*

**Response:** We have reorganized the section in the revised manuscript by clustering relevant contents together in line 165-175, as follows:

"$R_{NO_x:NO_2}$ is the ratio of $NO_x$ to $NO_2$. We use $R_{NO_x:NO_2}$ of 1.32 to represent "typical urban conditions and noontime sun" (Seinfeld and Pandis, 2006). We investigate the effect of using a constant value of $R_{NO_x:NO_2}$ on derived emissions in Sect. 3.1; it is found to be insignificant.

$b$ represents the $NO_2$ background for each city, which is derived by analyzing the distribution of $NO_2$ VCDs. We first calculate the mean $NO_2$ VCD under calm wind conditions for grid cells within the lowest 5th percentile of $NO_2$ VCDs for each city. This produces a good approximation of the mean $NO_2$ VCD for grid cells with low $NO_x$ emissions (i.e., the lowest 5th percentile of $NO_x$ emissions). We then multiply this mean VCD value by the spatial width of the across-wind integration interval to derive $b$.

$\tau$ is the $NO_x$ lifetime. Note that $\tau$ is assumed to be an effective mean dispersion lifetime (i.e., the result of the effect of deposition, chemical conversion, and wind advection) because we do not consider downwind changes in the fitting functions, such as due to variations in wind speeds or $R_{NO_x:NO_2}$ or lifetime itself."

*In the paragraph starting at Line 376 the authors discuss the uncertainties caused by clouds (which is good), but it's unclear if cloudy days were filtered out in the analysis. If not, then it would be important to mention this, perhaps near Line 188 of Section 2.2. Also, please mention that the $NO_2$ lifetime during a day with a true satellite observation will likely be smaller than the*

*values reported herein since sunny days yield faster photolysis rates.*

**Response:** Cloudy days are not excluded in the analysis. We have clarified this in Section 2.2, as follows:

"Note that we do not exclude cloudy days from our analysis to make the most of the NU-WRF $NO_2$ simulations and to avoid additional uncertainties arising from the inconsistent definitions of cloud fractions in the NU-WRF and satellite $NO_2$ products. The uncertainty of the presence of clouds is discussed in Sect. 3.3.2."

We also added the discussion about the bias of lifetime in Section 3.3.2, as follows:

"$NO_x$ lifetime on a sunny day with valid satellite observations will likely be shorter than that on a cloudy day since faster photolysis rates are expected for $NO_x$ reactions on sunny days."

*Line 285 - 288: r=0.01 is quite poor performance of the Beirle et al. 2011 method and a bit surprising. I wonder how the correlation would be if you eliminated "poor fitted" results (tau <1 hour and tau>5) in a similar manner to how certain cities were "eliminated" for the method described herein (Lines 238 - 241). It seems like the especially low correlation is driven by six outlier points, that if removed, might give better correlation. It'd be fair to do this if you are filtering out cities in your own method! This is not to say that the Beirle et al. method is equally good as the new method described herein, but it's probably not as bad as implied by the low correlation. I think it'd be fair to say that the Beirle et al. 2011 method might only work in a narrower range of cities (i.e., needs a stricter filter) as opposed to implying that it has almost no correlation in most circumstances.*

**Response:** We agree that the method of Beirle et al. (2011) works very well for isolated cities. For instance, it is capable of giving an accurate estimate for the isolated city of St. Louis in Missouri, with a relative difference of less than 10 % compared to the NU-WRF lifetime. We have calculated the value of *r* for the dataset excluding the 7 outlier points with fitted lifetime <1 h or > 5 h) and discussed it in the revised manuscript as follows:

"The correlation improves (R = 0.36) when eliminating the data for 7 cities with large (>5 h) or small (< 1h) fitted lifetimes, assuming the $NO_x$ emission distributions around these cities do not meet the requirements of the Beirle et al. method."

*Other minor suggestions:*
*Line 52: exploit —> use*

**Response:** We have replaced the term in the revised manuscript.

*Line 69: Perhaps mentioning validation field campaigns (e.g. DISCOVER-AQ, KORUS-AQ, CINDI-2) as being helpful to better quantify errors in the satellite data, and therefore reduce uncertainties in the applications of satellite data such as this project.*

**Response:** We have added this in the introduction, as follows:

"Field campaigns, e.g, Deriving Information on Surface Conditions from Column and Vertically Resolved Observations Relevant to Air Quality (DISCOVER-AQ), Korea–United States Air Quality Study (KORUS-AQ), and Cabauw Intercomparison of Nitrogen Dioxide Measuring Instruments 2 (CINDI-2), have been performed to better quantify errors in the $NO_2$ observations (e.g. Choi et al., 2020), and therefore improve knowledge about uncertainties in satellite-derived emissions."

*Figure 1: This figure is a bit confusing to me. I've attached an image in the supplement that is a bit more intuitive to me based on my understanding. Please feel free to discard if not correct or helpful.*

**Response:** Thanks for proposing such a useful diagram. We have replaced the figure following the comment in the revised manuscript, as follows:

[Figure]

**Figure 1 Schematic of our evaluation system to assess the accuracy of the inferring $NO_x$ lifetimes and emissions derived from MISATEAM. The blue symbols represent the inputs and outputs of MISATEAM. The orange symbols represent the information derived from NU-WRF.**

*Line 108: Please be more specific about how the emissions were adjusted. Were they projected*

*to a different year? If so, can you give a ballpark number as to how they are different than the 2011 NEI (10% lower? 20% lower? etc.)*

**Response:** We have added the details in the revised manuscript, as follows:

"We use the anthropogenic emissions based on the 2011 National Emissions Inventory (NEI) compiled by the US Environmental Protection Agency (US EPA, NEI 2011) but with a few modifications, in which the measurements from OMI, the ground-based Air Quality System (AQS), the in-situ continuous emissions monitoring in power plants, and the Air Pollutant Emissions Trends Data compiled by the US EPA (https://www.epa.gov/air-emissions-inventories/air-pollutant-emissions-trends-data), have been employed to adjust the baseline emissions to the simulation year of 2016 (Tong et al., 2015; Tao et al., 2020). As such, the total anthropogenic $NO_x$ emissions in 2016 were approximately 72% of those in 2011, the baseline NEI year."

*Line 114: Appears that the simulation is 6 months. Does it correspond to a specific 6 months in time? Presumably it is Apr - Sept, as implied in Line 159?*

**Response:** We ran the simulation for the whole year of 2016. But we only analyze data from April to September, in order to exclude winter data that have larger uncertainties and longer $NO_x$ lifetimes. We do use the data for other months to investigate the impact of the inclusion of winter data; it is found to be associated with a larger uncertainty. We have discussed this in Sect. 3.3.1, as follows:

"We also apply MISATEAM to year-round $NO_2$ data to investigate the impact of including winter data on the performance of the method. We keep default settings of MISATEAM as described in Sect. 2.2 for the fit. As expected, the fitted results differ more significantly from given values compared with results based on using only non-winter data. The bias is larger with NMB changing from 0.02 to -0.14 for lifetimes and from 0.13 to 0.27 for emissions. This indicates that MISATEAM, most likely its inherent steady-wind assumption, is less accurate during the winter season with longer $NO_x$ lifetimes."

*Line 114: Mention here that the domain is shown in Figure 2, and covers the Continental US.*

**Response:** We added this in the revised manuscript, as follows:

"Figure 2 illustrates the domain of the simulation, which covers the continental US."

*Line 115: This is nitpicky, but tropopause is closer to ~15 km during midsummer in most cases. How are you determining tropopause? From WRF or something else? Or are you assuming tropopause is consistently at the model top (50 hPa)?*

**Response:** We have performed a sensitivity analysis to identify an approximation to tropopause height for calculating tropospheric $NO_2$ VCD, because NU-WRF outputs do not include tropopause height directly and it is too expensive to calculate this information for each 4 km grid cells. We have used the tropopause information given in the Goddard Earth Observing System Model Version 5 (Pan et al., 2011) to develop the scenarios for sensitivity analysis. Since seasonal mean tropopause heights occur at altitudes between 8 (higher latitude or colder seasons) and 16 km (lower latitude or warmer seasons) over the US (Rieckh et al, 2014), we have integrated $NO_2$ concentrations from the surface to altitudes from 8 to 16 km. The derived mean $NO_2$ VCDs over the fit domain of individual cities vary very slightly, since most $NO_2$ stays near the surface over the polluted urban areas. We then assume a consistent tropopause height of 10 km to accelerate the data process. We have clarified this in the revised manuscript, as follows:
"We integrate $NO_2$ concentrations from the surface to the tropopause to calculate tropospheric $NO_2$ VCDs. We assume a consistent tropopause height of 10 km over the model domain to accelerate the data process, because NU-WRF outputs do not include tropopause height and $NO_2$ VCDs integrated above 10 km increase slightly. We have performed a sensitivity analysis by integrating $NO_2$ concentrations from the surface to altitudes ranging from 8 to 16 km, where the seasonal mean tropopause heights may occur over the US (Pan et al., 2011; Rieckh et al, 2014). The derived $NO_2$ VCDs over the fit domain of individual cities vary slightly above 10 km, with the relative difference of $3\% \pm 2\%$ when increasing the integration altitude from 10 to 12 km, since most $NO_2$ stays near the surface over the polluted urban areas."

*Line 127: Can you list 60 cities in the supplementary? Also can you briefly comment on why only 26 cities are shown on Figure 2? (I see a longer explanation in Section 3.1)*

**Response:** We have listed 60 cities in the revised supplementary material. We have clarified the reason for discarding cities from the final analysis in the revised manuscript, as follows:
"This filtering results in a total of 60 cities and urban conglomerations (see Table S1) as the candidates for applying MISATEAM, of which 26 have valid results. The locations of the 26

cities are shown in Figure 2. Cities without valid results either lack observations under calm wind conditions or are associated with large fitting errors (see details in Section 3.1)."

*Line 130: Would be helpful to mention here that a comparison to these two methods is forthcoming in Section 3.2.*

**Response:** We have added it in the revised manuscript, as follows:

"We develop a new model function aiming for determining emissions for mixed-sources, instead of isolated sources within a clean background considered by Beirle et al. (2011). It is also different from that of Liu et al. (2016b), which was developed for complex sources, but adapted an additional model function to fit emissions in a separate step. More comparisons with those two previous methods will be discussed in Sect. 3.2."

*Line 131: What does "Spatial emissions patterns" actually mean? This term is confusing to me. If I interpret Line 161 correctly, is it the instantaneous emissions rate (since the integral of it gives the emissions)?*

**Response:** It is the instantaneous emissions rate. We have changed the term to emission rates in the revised manuscript.

*Figure 4: Can you display the number of days included in each average? Presumably there are fewer days with easterly winds than westerly winds (for example).*

**Response:** We have updated the figure in the revised manuscript, as follows:

[Figure]

**Figure 4 NO₂ line densities around New York for different wind direction sectors. Circles: NO₂ line densities for calm (blue circles) and (A) southeasterly, (B) southerly, (C) southwesterly, (D) northeasterly, and (E) northerly winds (red circles) as a function of the distance *x* to New York center. Red line: the fit result *f(x)*. The numbers indicate the fitted NOₓ lifetime (τ), average days of data used for calculating NO₂ line densities (*Days*), derived emissions ($Emis_{NO_x}$) and given emissions ($Emis'_{NO_x}$). NO₂ line densities are derived from NO₂ VCDs averaged from April through September, 2016. NO₂ line densities for the remaining wind direction sectors are discarded due to the fitting results having insufficient quality.**

*Figure 5 (and S2): I very much like these figures, but wonder what information is missing by rounding the speed to 5 knots. Could you display the mean wind speed at each hour in a box on the bottom right of each of the nine panels?*

**Response:** We have added the mean wind speed in the figures in the revised manuscript and supplement. To better address the next comment, we select the direction with smallest and largest fit errors for illustration.

[Figure]

**Figure 5 Wind barbs around New York City for different times of the day. All northeasterly winds at 14:00 local time from April to September of 2016 are averaged and shown in (I). Wind barbs for the northeasterly winds backward trajectories from 8 to 1 h prior to 14:00 local time are displayed in (A) – (H). Wind speed is given in the units of knots, which is a nautical miles per hour (1.9 km per hour). Each short and long barb represents 5 knots (9.3 km/h) and 10 knots (18.5 km/h), respectively. The average wind speed is displayed in the grey box.**

[Figure]

**Figure S3 Similar to Figure 5, but for westerly wind.**

*Line 201: This sentence appears to be a bit contradictory with Line 202. For New York City, easterly winds don't do as well, but Line 201 seems to imply that all wind directions are OK. This should be re-phrased. More generally though, is there worse performance with easterly winds? That would be interesting information for the reader to know.*

**Response:** We are not able to get valid fit results for all directions. We thus discard the direction without good fit results from the analysis. We have added the figures for rejected directions in the Figure S2 of the revised supplementary, as follows:

[Figure]

**Figure S2 Similar to Figure 5, but for wind direction sectors with fitting results of insufficient quality: (A) westerly and (B) northwesterly wind.**

We have rephrased the sentence in the revised manuscript, as follows:

"We use the northeasterly wind direction (with a good fitting result) for demonstration. We select northeasterly winds observed at 14:00 local time as the baseline and find their backward trajectories for up to 8 hours. The backward trajectories are given at a time step of one hour. Not surprisingly, winds are not constant during the 9 hours from 8 hours before the baseline to the exact hour of the baseline. However, the temporal variations in wind directions are rather small for the northeasterly wind; wind directions are almost constant over time. For wind directions without good fit results, we observe larger variations. For instance, for the westerly wind with a poor correlation coefficient $R$ of 0.76, the wind directions deviate from the west direction gradually for the time prior to the baseline (Fig. S3)."

*Line 255: I think you mean that there is "increasing agreement" over the 1 hr lifetime. Might also want to note that tau=3,6,9 are all similar, and that tau=12 does seem worse then 3,6,9, but better than 1 h (and this make sense since -12 h includes overnight!)*

**Response:** We have rephrased the sentence in the revised manuscript, as follows:

"The use of wind information prior to the satellite overpass time succeeds in improving the performance of MISATEAM in all these cases (Fig. S4). Note that the correlation between the inferred and the NU-WRF lifetimes based on 12 h wind (R=0.64) is not as good as that based on 3 h (R=0.74), 6 h (R=0.78), and 9 h (R=0.79) wind, which is most likely caused by the inclusion of overnight wind information."

*Line 301: Might be better to re-phrase to say that MISATEAM is better than Liu et al.,2016*

**Response:** We have rephrased the sentence in the revised manuscript, as follows:

"We note that the performance of MISATEAM is also better than that of the approach reported in Liu et al. (2016b)".

*Line 346: "layer height" —> "top wind layer height" or ""wind layer depth"*
**Response:** We have corrected the term to top wind layer height in the revised manuscript.

*Line 392: It's probably worthwhile to also mention these uncertainties in the abstract (Line 27) and conclusions (Line 413) in addition to the 15% / 20% values currently mentioned.*
**Response:** We have added the discussion about the uncertainties in the abstract and conclusion, as follows:

abstract: "The total uncertainties reach up to 43 % (lifetimes) and 45 % (emissions) by considering the additional uncertainties associated with satellite $NO_2$ observations and wind data."

conclusion: "Additional uncertainties are associated with wind errors in the reanalysis dataset as well as errors in the satellite $NO_2$ retrievals, increasing the total uncertainties of $NO_x$ lifetime and emissions to 43 % and 45 %, respectively."

*Line 411: Maybe add a short comment to say that diurnal $NO_2$ lifetime differences will need to be investigated before applying this method to hours outside of the early afternoon timeframe. Or perhaps you are even willing to say that the method can only be applied in the early afternoon based on work applying this method to timeframes with longer $NO_2$ lifetimes (Line 354)*
**Response:** We have added the discussion about diurnal $NO_2$ lifetime in the revised manuscript, as follows:

"For applications based on geostationary satellites with local observation time outside of the early afternoon time frame, additional investigation about the impact of the diurnal cycle of $NO_2$ lifetime is required, since MISATEAM is expected to have a larger uncertainty when the lifetime is longer."

*Line 414: Maybe be more explicit and mention a current low bias in TROPOMI $NO_2$ (Verhoelst*

*et al. 2021) that would likely yield low satellite-derived NOx emissions if not bias-corrected.*

**Response:** We have added the discussion about the bias of the TROPOMI NO$_2$ products in the revised manuscript, as follows:

"The general low bias of NO$_2$ Tropospheric VCDs from TROPOMI for polluted sites (Verhoelst et al. 2021) is directly transferred into the inferred NO$_x$ emissions if no correction is performed."

*Please also note the supplement to this comment:*

*https://acp.copernicus.org/preprints/acp-2021-642/acp-2021-642-RC1-supplement.pdf*

**Response:** Thanks again for providing such a detailed diagram to help us improve the manuscript. We have revised Figure 1 following the suggestion.

Reference

Choi, S., Lamsal, L. N., Follette-Cook, M., Joiner, J., Krotkov, N. A., Swartz, W. H., Pickering, K. E., Loughner, C. P., Appel, W., Pfister, G., Saide, P. E., Cohen, R. C., Weinheimer, A. J., and Herman, J. R.: Assessment of NO$_2$ observations during DISCOVER-AQ and KORUS-AQ field campaigns, Atmos. Meas. Tech., 13, 2523–2546, https://doi.org/10.5194/amt-13-2523-2020, 2020.

Pan, L. L., and L. A. Munchak, Relationship of cloud top to the tropopause and jet structure from CALIPSO data, J. Geophys. Res., 116, D12201, doi:10.1029/2010JD015462, 2011.

Rieckh, T., Scherllin-Pirscher, B., Ladstädter, F., and Foelsche, U.: Characteristics of tropopause parameters as observed with GPS radio occultation, Atmos. Meas. Tech., 7, 3947–3958, https://doi.org/10.5194/amt-7-3947-2014, 2014.

Verhoelst, T., Compernolle, S., Pinardi, G., Lambert, J.-C., Eskes, H. J., Eichmann, K.-U., Fjæraa, A. M., Granville, J., Niemeijer, S., Cede, A., Tiefengraber, M., Hendrick, F., Pazmiño, A., Bais, A., Bazureau, A., Boersma, K. F., Bognar, K., Dehn, A., Donner, S., Elokhov, A., Gebetsberger, M., Goutail, F., Grutter de la Mora, M., Gruzdev, A., Gratsea, M., Hansen, G. H., Irie, H., Jepsen, N., Kanaya, Y., Karagkiozidis, D., Kivi, R., Kreher, K., Levelt, P. F., Liu, C., Müller, M., Navarro Comas, M., Piters, A. J. M., Pommereau, J.-P., Portafaix, T., Prados-Roman, C., Puentedura, O., Querel, R., Remmers, J., Richter, A., Rimmer, J., Rivera Cárdenas, C., Saavedra de Miguel, L., Sinyakov, V. P., Stremme, W., Strong, K., Van Roozendael, M., Veefkind, J. P., Wagner, T., Wittrock, F., Yela González, M., and Zehner, C.: Ground-based validation of the Copernicus Sentinel-5P TROPOMI NO$_2$ measurements with the NDACC ZSL-DOAS, MAX-DOAS and Pandonia global networks, Atmos. Meas. Tech., 14, 481–510, https://doi.org/10.5194/amt-14-481-2021, 2021.

Anonymous Referee #2

*This paper describes an improved method for estimating emissions from NO₂ columns. The method is applied to model results so that it can be evaluated thoroughly, with a view to application to satellite data in the future.*

*The paper is interesting and well written, and the subject is significant. I am happy to recommend publication.*

**Response:** We thank Referee #2 for the encouraging comments. We addressed all the comments carefully as below.

*My main comment is that the method is presented as being "new" but is really an evolution of prior work. This is not a problem, but I think the presentation would be greatly helped by describing more clearly what the method was before, and what the change is. I know that the details are already in the manuscript, but at the moment it is a little difficult to follow and to figure out what the significant changes are.*

*Given that the paper is a methods paper, it would help if the figures and description were more pedagogical in nature. They could show more clearly what the method does and how. This would help the reader follow the explanation.*

**Response:** In order to make it more clearly what the method does and how,

We have updated the schematic of the work in Figure 1 based on the comments from reviewer #1, as follows:

[Figure]

**Figure 1 Schematic of our evaluation system to assess the accuracy of the inferring NOₓ lifetimes and emissions derived from MISATEAM. The blue symbols represent the inputs (dash line) and outputs (solid line) of MISATEAM. The orange symbols represent the information derived from NU-WRF.**

We have also added a flow chart to clarify the changes of this study compared to previous work in Figure S5 of the revised supplementary, as follows:

[Figure]

**Figure S5 Comparison of methodology between this study (MISATEAM) and Beirle et al. (2011) and Liu et al. (2016b). *MISATEAM and the approach of Liu et al. (2016b) are also applicable to a single point source.**

*The study uses 26 cities but does not show much about the difference between them – we barely see where these cities are. Some more details should be presented, at least in the SI if not in the main text.*

**Response:** We have listed the details of the cities in the table S1 of the revised supplementary.

*I think it would be informative to say what percentage of the days are in each wind direction sector. It would also be interesting to see some of the rejected sectors – for someone wanting to replicate the study, it would be helpful to see what was kept and what was rejected. The text could briefly explain the rational for the decision.*

**Response:** We added the number of days used for each wind direction sector in the revised Figure 4, as follows:

[Figure]

**Figure 4 NO₂ line densities around New York for different wind direction sectors. Circles: NO₂ line densities for calm (blue circles) and (A) southeasterly, (B) southerly, (C) southwesterly, (D) northeasterly, and (E) northerly winds (red circles) as a function of the distance *x* to New York center. Red line: the fit result *f(x)*. The numbers indicate the fitted NOₓ lifetime ($\tau$), average days of data used for calculating NO₂ line densities (*Days*), derived emissions ($Emis_{NO_x}$) and given emissions ($Emis'_{NO_x}$). NO₂ line densities are derived from NO₂ VCDs averaged from April through September, 2016. NO₂ line densities for the remaining wind direction sectors are discarded due to the fitting results being of insufficient quality.**

We have also added the figures for rejected directions in the Figure S4 of the revised supplementary, as follows:

[Figure]

**Figure S4 Similar to Figure 4, but for wind direction sectors with fitting results of insufficient quality: (A) westerly and (B) northwesterly wind.**

We have explained the reason for the rejection in the manuscript, as follows:

"Fitting results of insufficient quality (i.e., the correlation coefficient $R$ between the fitted and observed $NO_2$ LD < 0.9, and one standard deviation error of $\tau$ > 10%) are discarded."

*I thought Fig 5 was an odd choice to include in the manuscript. It seems like typical SI stuff (if that even). Maybe moving it would free up some space to include more figures describing the method and the results themselves.*

**Response:** We thank you for the suggestions and fully understand the concerns here. However, since Reviewer 1 indicated that they considered this an important figure, we decided to keep it.

*Nomenclature was sometimes a bit clumsy. "ratio" and "emis' "could have better names for clarity and legibility.*

**Response:** We have changed *ratio* to $R_{NO_x:NO_2}$ and *emis* to $Emis_{NO_x}$ in the revised manuscript.